# *Bacillus velezensis* Y6, a Potential and Efficient Biocontrol Agent in Control of Rice Sheath Blight Caused by *Rhizoctonia solani*

**DOI:** 10.3390/microorganisms12081694

**Published:** 2024-08-16

**Authors:** Huan Tao, Xiaoyu Li, Huazhen Huo, Yanfei Cai, Aihua Cai

**Affiliations:** 1Guangxi Key Laboratory of Plant Functional Phytochemicals and Sustainable Utilization, Guangxi Institute of Botany, Guangxi Zhuang Autonomous Region and Chinese Academy of Sciences, Guilin 541006, China; taohuanscau@163.com (H.T.); 18978363118@163.com (H.H.); 2College of Natural Resources and Environment, South China Agricultural University, Guangzhou 510462, China; xiaoyu1999@stu.scau.edu.cn

**Keywords:** rice sheath blight, *Bacillus velezensis*, biocontrol, lipopeptides, transcriptome

## Abstract

Rice sheath blight is a serious disease caused by *Rhizoctonia solani* that reduces rice yield. Currently, there is a lack of efficient and environmentally friendly control methods. In this study, we found that *Bacillus velezensis* (*B. velezensis*) Y6 could significantly inhibit the growth of mycelium in *Rhizoctonia solani*, and its control efficiency against rice sheath blight was 58.67% (*p* < 0.01) in a pot experiment. Lipopeptides play an important role in the control of rice sheath blight by *B. velezensis* Y6, among which iturin and fengycin are essential, and iturin W, a novel lipopeptide in *B. velezensis*, plays a major role in lipopeptide antagonism to *Rhizoctonia solani*. In the field, we also found that inoculation with *B. velezensis* Y6 can increase rice yield (dry weight) by 11.75%. Furthermore, the transcriptome profiling results of the rice roots revealed that there were a total of 1227 differential genes (DEGs) regulated when treated with Y6, of which 468 genes were up-regulated and 971 genes were down-regulated in rice roots compared with the control. Among them, the DEGs were mainly distributed in biological processes (BP) and were mainly enriched in response to stimulus (GO:0050896), response to stress (GO:0006950), and response to abiotic stimulus (GO:0009628). According to the KEGG pathway analysis, there were 338 DEGs classified into 87 KEGG functional pathway categories. Compared with the control, a large number of enriched genes were distributed in phenylpropanoid biosynthesis (map00940), glutathione metabolism (map00480), glycolysis/gluconeogenesis (map00010), and amino sugar and nucleotide sugar metabolism (map00520). In summary, this investigation provides a new perspective for studying the molecular mechanism of *B. velezensis* in controlling rice sheath blight.

## 1. Introduction

Rice sheath blight is one of China’s three major rice diseases [1]. The disease is caused by *Rhizoctonia solani* Kühn and has a wide disease area and a high frequency of incidence, which brings huge obstacles to the high and stable yield of rice [2]. At present, there are no cultivars with high resistance to rice sheath blight in rice cultivation, and agrochemical application is still the main means of controlling this disease [3,4]. However, the extensive use of pesticides in the field can lead to increased disease resistance and environmental pollution, and the search for safer and more effective methods of controlling rice sheath blight is now a mainstream trend. Biological control is widely recognized as a reliable, safe, and sustainable method of plant disease control [5,6].

Biological control mainly involves the use of microbial strains such as *Bacillus*, *Harzianum Pseudomonas*, *Actinomycetes* and *Burkholderia* to control plant diseases [7]. The use of *Bacillus* or its metabolites to control plant diseases is currently the focus of biocontrol research [8,9]. *Bacillus* can act as a biocontrol agent against fungal, bacterial and viral infections by inducing systemic resistance (ISR) in host plants [10,11]. *B. velezensis* YYC enhances the activity of defense-related enzymes (e.g., PAL, POD, and SOD), thereby increasing the basal immunity of the plant [12]. *Bacillus cereus* (Bce-2) prevents strawberry leaf spot disease caused by *N. clavispora* via increasing the expression of disease resistance genes in strawberry leaves [13]. *Bacillus cereus* AR156 can induce systemic resistance of *Arabidopsis thaliana* to *Botrytis cinerea* via JA/ET and NPR1-dependent signaling pathways [14]. In addition, *Bacillus* produces a variety of antibiotics, such as cyclic lipopeptides (CLPs), which can directly antagonize pathogens [15,16]. Surfactants secreted by *Bacillus amyloliquefaciens* Ba01 play an important role in the biological control of potato common scab [17]. Iturin A and Fengycin A secreted by *Bacillus megaterium* WL-3 have an inhibitory effect on *P. infestans* mycelium growth, as well as a good control effect on potato late blight in greenhouse experiments and field assays [18]. Although there are many studies on the biological control of plant diseases by *Bacillus*, the application of *B. velezensis* to the biological control of rice blight has not been reported.

In this study, we verified through pot experiments that *B. velezensis* Y6 could effectively control rice sheath blight. The main types and basic structures of lipopeptides secreted by *B. velezensis* Y6 were discovered by mass spectrometry; the lipopeptides essential for the antagonism of *Rhizoctonia solani* by *B. velezensis* Y6 were discovered by constructing mutant strains with deletions of genes related to lipopeptide synthesis; and the effect of *B. velezensis* Y6 on the expression of the rice transcriptome was revealed by RNAseq. This study provides new insights into how *B. velezensis* could be used to control rice sheath blight.

## 2. Materials and Methods

### 2.1. Strains and Growth Media

The strains used in this study are listed in Appendix A. Mutants lacking lipopeptide synthesis were constructed as described previously [19]. *B. velezensis* Y6 and its derivative mutant strains were cultured on LB (10 g of tryptone, 5 g of yeast extract, and 5 g of NaCl, and distilled water 1 L at pH 7.0–7.2), or on solid LB medium supplemented with 1.5% agar. The corresponding antibiotics were added when culturing mutants. *Rhizoctonia solani* AG1-IA GD118 was cultured on PDA (potato 200 g/L, glucose 20 g/L, agar 20 g/L, and distilled water 1 L at pH 7.0–7.2) at 30 °C. The rice variety was Meixiangzhan 2.

### 2.2. B. velezensis Y6 Antagonizes Rhizoctonia solani

*Rhizoctonia solani* AG1-IA GD118 was cultured on PDA plates at 30 °C for 3 days. A 5 mm diameter mycelial agar block was cut and transferred to the center of a new PDA plate. After 1 day of culture, 30 µL of *B. velezensis* Y6 or its derivative mutants (OD600 ~0.4) grown in LB medium was added to a 5 mm diameter well 2.5 cm from the center of the mycelial agar block. Antifungal activity was evaluated by measuring the diameter of the inhibition zone after 2 days of culture at 30 °C.

### 2.3. Pot Experiment

After soaking rice seeds in warm water for 1 day, they were germinated at 30 °C for 2 days before sowing. When the rice had 3 leaves at 28 °C, it was transplanted. The *B. velezensis* Y6 solution with a concentration of 1 × 10^6^ CFU/mL was inoculated on the rice root. After overnight cultivation, the rice was transplanted into a 6 L planting bucket (containing 5 kg of paddy soil). At 15 days after transplanting, the Y6 liquid bacterial agent with a concentration of 10^6^ CFU/mL was sprayed on the rice until the rice leaves were fully moistened. The control group was sprayed with water, and the pesticide group was sprayed with 5% of Validamycin. At 24 h after the treatment, the *Rhizoctonia solani* grown on 2% agar PDA medium was cut into 5 mm pieces, fixed on the leaf sheath of the lowest rice leaf with plastic wrap, and kept moist with absorbent cotton. The disease severity data were collected from 15 days after the pathogen inoculation. The degree of disease severity was assigned as follows (Figure 6A): 0 = no lesions, 1 = appearance of water-soaked lesions, 2 = appearance of necrotic lesions, 3 = less than 50% necrosis on the leaf cross section, 4 = more than 50% necrosis on the leaf cross section, and 5 = necrosis across the entire leaf section resulting in leaf death [4]. The severity of the disease and the biocontrol efficiency were calculated according to the following formula.
$$\mathrm{Disease}\;\mathrm{severity}=\frac{\sum (Number\;of\;diseased\;plants\times Disease\;level)}{\left(Total\;number\;of\;plants\times Maximum\;disease\;level\right)}$$
$$\mathrm{Control}\;\mathrm{efficiency}=\frac{Control\;disease\;severity-Treatment\;disease\;severity}{Control\;disease\;severity}$$

### 2.4. Field Experiment

The field experiment was conducted on 1 April 2022, at the South China Agricultural Teaching and Research Base, Zengcheng District, Guangzhou City, Guangdong Province (113.633677° E, 23.243212° N, 22 m a.s.l.). The rice variety used was Meixiangzhan No. 2 (conventional rice), with a planting area of 30 m^2^ in each plot, a plant spacing of 15 cm, a row spacing of 30 cm, and a spacing of 50 cm between plots. Each treatment was repeated four times.

The rice seedlings were inoculated with *B. velezensis* Y6 solution at a concentration of 1 × 10^6^ CFU/mL by root irrigation one day before transplanting. The field was managed with normal fertilizer and water, and *B. velezensis* Y6 solution with a concentration of 1 × 10^6^ CFU/mL was sprayed on the leaves during the rice booting stage. After the rice matured, 20 rice plants were randomly selected from each plot to determine the number of effective panicles. The rice yield was determined by sampling and measuring the yield in the plot. The yield measurement area was 1 m^2^, and the measurement was repeated three times along the diagonal of the plot.

### 2.5. Extraction of Crude Lipopeptides

Crude lipopeptides were extracted by acid precipitation and methanol extraction. *B. velezensis* Y6 and its derivative mutant strains were cultured on LB liquid medium at 37 °C for 1 d. The culture was centrifuged at 12,000 rpm for 10 min to remove cells; the supernatant was adjusted to pH 2.0 with concentrated hydrochloric acid and allowed to precipitate overnight. It was then centrifuged at 12,000 rpm for 10 min and the supernatant was discarded. The precipitate was retained and extracted twice with 1/10 volume of methanol. It was allowed to stand at 4 °C for 2 h each time, and then the methanol extracts were combined.

### 2.6. Extraction of Lipopeptides (LP) from the Transparent Inhibition Zone

We extracted LPs from the clear zone (Figure 1A). A 600 mg agar sample was collected from the clear zone, minced and mixed with 2 mL of acetonitrile/water (1:1 *v*/*v*), shaken, sonicated twice for 30 s each and centrifuged, and then the supernatant was collected and filtered [19].

### 2.7. Identification of LPs by UPLC/Q-TOF-MS

The acetonitrile/water extracts were analyzed by reversed-phase Ultra-Performance Liquid Chromatography coupled with Time-of-Flight Mass Spectrometry (UPLC/Q-TOF-MS). The mass-to-charge ratio (*m*/*z*) was used to identify the lipopeptide compounds. Gradient elution was performed using (A) acetonitrile and (B) water (containing 0.2% formic acid), and the column temperature was maintained at 40 °C [20]. The detection conditions of the Agilent Ultra-Performance Liquid Chromatography coupled with Time-of-Flight Mass Spectrometry (UPLC/Q-TOF-MS) were as shown in Table 1:

### 2.8. Total RNA Extraction and Transcriptomic Analysis

To evaluate the effect of Y6 on the rice transcriptome, the following experiment was performed: 15 days after rice plants were transplanted, three samples were collected from the roots of rice plants in the control and Y6 treatments. The rice roots were washed twice with sterile water and wiped dry, then quickly frozen in liquid nitrogen, and finally the samples were stored in an ultra-low-temperature freezer at −80 °C. Total RNA was then extracted using Trizol reagent (Invitrogen Life Technologies, Carlsbad, CA, USA). The concentration, purity, and integrity of the RNA were then assessed using a NanoDrop spectrophotometer (Thermo Scientific, Waltham, MA, USA).

We selected a total amount of >=1 μg of RNA and used the NEBNext Ultra II RNA Library Prep Kit for Illumina to construct the library. We used the Agilent 2100 Bioanalyzer (Agilent, Santa Clara, CA, USA) and the Agilent High Sensitivity DNA Kit (Agilent, 5067-4626) to determine the library quality. We used Cutadapt (v1.15) software to filter the sequencing data and obtain high-quality sequences (clean data) for further analysis. The Oryza sativa genome was constructed using IRGSP-1.0 as the reference genome. A term was assigned to each gene in the Gene Ontology database, and the number of differentially enriched genes in each term was calculated. The hypergeometric distribution method was used to calculate the *p* value, and topGO (2.40.0) was used to perform GO enrichment analysis on the differentially expressed genes (the benchmark for significant enrichment was *p* value 0.05) and locate the GO of significant enrichment of the differentially expressed genes to determine the main biological functions of the differentially expressed genes. ClusterProfiler (3.16.1) software was used to perform enrichment analysis on the KEGG pathways of differentially expressed genes, with an emphasis on the significantly enriched pathways.

### 2.9. Statistical Analysis

Statistical analysis in the experiments was performed using the *t* test in IBM SPSS Statistics 20, and *p* < 0.05 indicated a significant difference. Tables were created using GraphPad Prism 8.

## 3. Results

### 3.1. B. velezensis Y6 Can Effectively Inhibit the Growth of Rhizoctonia solani

A plate antagonism test was used to investigate *B. velezensis* Y6’s antagonistic effect on the rice sheath blight pathogen. *Rhizoctonia solani*’s mycelium can be inhibited by *B. velezensis* Y6, as shown in Figure 1A, resulting in the formation of a clear, transparent region on the plate. The lipopeptides secreted by *B. velezensis* Y6 were extracted, and the plate dual culture method was used to study the effect of the lipopeptides secreted by *B. velezensis* Y6 on rice sheath blight pathogen. As shown in Figure 1B, *B. velezensis* Y6 can significantly restrict the growth of mycelium of *Rhizoctonia solani*. The above results show that *B. velezensis* Y6 can effectively antagonize rice sheath blight pathogens, and lipopeptides play an important role in this process.

### 3.2. Characterization of B. velezensis Y6-Secreted Lipopeptides by UPLC/Q-TOF-MS

To clarify the types and structures of lipopeptides (LPs) produced by *B. velezensis* Y6, we extracted LPs from the transparent inhibition zone using acetonitrile/water (1:1) and then analyzed them by UPLC/Q-TOF-MS. According to the mass-to-charge ratio (*m*/*z*), we found three types of compounds in the extracted lipopeptides: iturin, fengycin, and surfactin (Appendix A). Each LP has multiple homologues, depending on the length of the fatty acid carbon chain. According to the chromatographic peak retention time, we found that iturin has four homologues (fatty acid carbon chain length C14 to C17), fengycin has four homologues (fatty acid carbon chain length C14 to C15), and surfactin has five homologues (fatty acid carbon chain length C12 to C16) (Appendix A). Iturin, fengycin, and surfactin all have a ring structure composed of amino acid chains and fatty acids. In order to determine the molecular structure of the compounds, the information from their secondary mass spectra was collected and analyzed. Figure 2 is an ion fragment diagram formed by the cleavage of iturin with a mass-to-charge ratio (*m*/*z*) of 1043.5551, which mainly includes b-type ion fragments generated from the cleavage of the N-terminus and y-type ion fragments generated from the cleavage of the C-terminus. Starting from the N-terminus, the b-ion fragments were 915.49 (b7), 801.44 (b6), 638.39 (b5), 541.25 (b4), 427.29 (b3), and 340.26 (b2). Assuming that the value of [M+H]^+^ is 1043.5551, the differences between these values were exactly the masses of Gln, Asn, Tyr, Pro, Asn, Ser, and Asn ion fragments. Starting from the C-terminus, the detectable y-ion fragments were 704.29 (y6), 617.37 (y5), 503.30 (y4), 406.16 (y3), and 243.11 (y2). The differences between their values corresponded to the masses of Ser, Asn, Pro, and Tyr ion fragments, which is consistent with the analysis results of the b-ion fragments. The preliminary amino acid sequence of iturin (1043.5551) was speculated to be β-amino fatty acid-Asn-Ser-Asn-Pro-Tyr-Asn-Gln from the N-terminus to the C-terminus, which belongs to Iturin W in the iturin family. Figure 3 is an ion fragment diagram formed by the cleavage of surfactin with a mass-to-charge ratio (*m*/*z*) of 1036.6921, which mainly includes b-type ion fragments generated from the cleavage of the N-terminus and y-type io1036.6921n fragments generated from the cleavage of the C-terminus. Starting from the N-terminus, the b-ion fragments were 923.61 (b7), 810.60 (b6), 695.45 (b5), 596.42 (b4), 483.34 (b3), and 370.07 (b2). Assuming that the value of [M+H]^+^ is 1036.6921, the differences between these values were exactly the masses of Leu, Leu, Asp, Val, Leu, Leu, and Glu ion fragments. Starting from the C-terminus, the detectable y-ion fragments were 667.43 (y6), 554.35 (y5), 441.27 (y4), 342.23 (y3), and 227.17 (y2). The differences between their values were the masses of Leu, Leu, Val, and Asp ion fragments, which is consistent with the analysis results of the b-ion fragments. The preliminary amino acid sequence of surfactin (1036.6921) was speculated to be β-amino fatty acid- Glu-Leu-Leu-Val-Asp-Leu-Leu from the N-terminus to the C-terminus, which belongs to Surfactin A in the surfactin family. Figure 4 is an ion fragment diagram formed by the cleavage of fengyrin with a mass-to-charge ratio (*m*/*z*) of 1463.8120 and 1491.8505. In the primary structure of fengycin, the main difference is the amino acid composition at the sixth position, which is usually Ala or Val. According to the characteristic ion fragment mass-to-charge ratio of 1080.5380 and 996.4575, it is judged that the sixth position of the amino acid chain is Ala, belonging to Fengycin A. According to the characteristic ion fragment mass-to-charge ratios of 1108.5701 and 994.4948, it is judged that the sixth position of the amino acid chain is Val, belonging to Fengycin B. These results show that Fengycin secreted by *B. velezensis* Y6 contains two structures: Fengycin A and Fengycin B.

### 3.3. Antibacterial Activity of B. velezensis Y6 and Its Derived Mutants against Rhizoctonia solani

In order to clarify the type of lipopeptides that play a major role in the antagonism of *B. velezensis* Y6 to rice sheath blight, the effects of the wild-type and three mutants (453 (*srfAA::mls*), 454 (*ituA::mls*), and 459 (*fenC::spc*)) of *B. velezensis* Y6 on the mycelial growth of *Rhizoctonia solani* were studied by the plate dual-culture method. The results in Figure 5 show that there were obvious differences in the antibacterial activity of the wild type and mutant strains of *B. velezensis* Y6 against the *Rhizoctonia solani*. The diameter of the inhibition zone of the wild type of *B. velezensis* Y6 was 24.88 ± 0.22 mm. Compared with the wild type of *B. velezensis* Y6, the inhibition zone diameter of mutant strain 453 (*srfAA::mls*) was 24.50 ± 0.50 mm, with no significant difference; the inhibition zone diameter of mutant 454 (*ituA::mls*) was 17.13 ± 0.74 mm, which was reduced by 31.15%, with a significant difference (*p* < 0.01); the inhibition zone diameter of mutant 459 (*fenC::spc*) was 21.88 ± 0.65 mm, which was reduced by 12.06%, with a significant difference (*p* < 0.01) (Table 2). These results indicate that fengycin and iturin are essential for the antagonism of *B. velezensis* Y6 against the *Rhizoctonia solani*, among which iturin plays a major role in the antagonism of *B. velezensis* Y6 against the *Rhizoctonia solani*.

### 3.4. Biocontrol Effects of Y6 on Rice Plant Tissues in Pot Experiment

We found that *B. velezensis* Y6 can effectively inhibit the growth of *Rhizoctonia solani.* In order to evaluate the effect of *B. velezensis* Y6 on the biological control of rice sheath blight, we conducted a pot experiment. As shown in Figure 6, the disease severity of rice sheath blight in the control group was 83.33 ± 5.98%, and the disease severity of rice sheath blight in the group treated with 5% Validamycin was 17.22 ± 3.29%. Compared with the control group, the disease severity of rice sheath blight in the group inoculated with *B. velezensis* Y6 was 34.44 ± 8.39%, and the control efficiency was 58.67% (*p* < 0.01). The above results show that *B. velezensis* Y6 can be an effective biological control of rice sheath blight.

### 3.5. B. velezensis Y6 Can Increase Rice Yield in the Field

In general, *B. velezensis* has a growth-promoting effect on plants. In order to verify the effect of *B. velezensis* Y6 on rice yield, we conducted pot experiments and field experiments. In the rice pot experiment (Appendix A), we found that after inoculation with *B. velezensis* Y6, the thousand-grain weight, seed-setting rate, and effective panicle number of rice increased by 5.4%, 22.18% (*p* < 0.01), and 21.23% (*p* < 0.01), respectively (Appendix A). Through field experiments, we confirmed the effect of *B. velezensis* Y6 on rice yield (Appendix A). The results showed that, compared with the control group, the panicle length, 1000-grain weight, and rice yield (dry weight) of rice increased by 7.7% (*p* < 0.05), 5.4%, and 11.75%, respectively, after inoculation with *B. velezensis* Y6 (Appendix A). The above results show that inoculation with *B. velezensis* Y6 can increase rice field yield.

### 3.6. Transcriptome Profiling Effects of Strain Y6 on Rice

We analyzed the transcriptome of rice roots inoculated with strain Y6. The results showed that 468 genes were up-regulated and 971 genes were down-regulated in rice roots compared with the control (Figure 7). In addition, we also performed enrichment analysis of DEGs in rice using the GO database. A total of 1439 genes were enriched in 167 categories of three major categories in the GO database: molecular function (MF), cellular component (CC), and biological process (BP). The top 20 GO entries with the most significant enrichment were selected in the three broad categories of CC, MF, and BP (Appendix A). The DEGs in CC were mainly enriched in the external encapsulating structure (GO:0030312), cell wall (GO:0005618), cell periphery (GO:0071944), extracellular region (GO:0005576), and vacuole (GO:0005773) categories. In MF, they were mainly enriched in the signaling receptor binding (GO:0005102), cytoskeletal motor activity (GO:0003774), and transporter activity (GO:0005215) categories. In BP, they were mainly enriched in the response to stimulus (GO:0050896), response to stress (GO:0006950), response to abiotic stimulus (GO:0009628), carbohydrate metabolic process (GO:0005975), cellular homeostasis (GO:0019725), homeostatic process (GO:0042592), regulation of biological quality (GO:0065008), response to biotic stimulus (GO:0009607), transport (GO:0006810), localization (GO:0051179), establishment of localization (GO:0051234), and lipid metabolic process (GO:0006629) categories. While in the GO factor graph, DEGs were mainly distributed in BP and were mainly enriched in the response to stimulus (GO:0050896), response to stress (GO:0006950), and response to abiotic stimulus (GO:0009628) categories (Figure 8).

In addition, we performed enrichment analysis of DEGs in rice using the KEGG database. There were 338 DEGs classified into 87 KEGG functional pathway categories. The 10 most significant pathways mainly involved in plant metabolic pathways were phenylpropanoid biosynthesis (map00940), glutathione metabolism (map00480), glycolysis/gluconeogenesis (map00010), amino sugar and nucleotide sugar metabolism (map00520), starch and sucrose metabolism (map00500), valine, leucine and isoleucine degradation (map00280), cysteine and methionine metabolism (map00270), terpenoid backbone biosynthesis (map00900), butanoate metabolism (map00650), and circadian rhythm—plant (map04712) (Appendix A). In addition, a large number of enriched genes were distributed in phenylpropanoid biosynthesis (map00940), glutathione metabolism (map00480), glycolysis/gluconeogenesis (map00010), amino sugar and nucleotide sugar metabolism (map00520) (Figure 9).

## 4. Discussion

This study aimed to elucidate the mechanism of *B. velezensis* Y6 in controlling rice sheath blight. Studies have found that strain Y6 can antagonize rice sheath blight pathogens, in which the lipopeptides play an important role. Pot experiment results show that inoculation with strain Y6 can significantly reduce the disease severity of rice sheath blight.

The antibacterial activity of *Bacillus* is related to the lipopeptides it produces [21,22]. *Bacillus* strains can create one or more types of lipopeptides, each of which usually includes numerous homologues with identical amino acid residues and various lengths of fatty acid chains [23,24]. Lipopeptides belonging to *B. velezensis* Y6 were characterized by UPLC/Q-TOF-MS analysis, and three subfamilies, Surfactin A (C12–C16), Iturin W (C14–C17), Fengycin A and Fengycin B (C16–C17) subfamilies, were found. Zhou et al. reported that Iturin W, a novel lipopeptide produced by the deep-sea bacterium *Bacillus* sp. strain wsm-1, has stronger antibacterial activity than the common iturin A and has good prospects for biological control of rice blast disease caused by *M. grisea* [25]. This is the first time that iturin W has been reported in *B. velezensis* and it plays an important role in the biological control of rice sheath blight by *B. velezensis* Y6. Surfactin, iturin, and fengycin generally have strong antagonistic activities against pathogens; however, their activities against *Rhizoctonia solani* were still unclear. By constructing lipopeptide synthesis-related gene deletion mutants (453 (*srfAA::mls*), 454 (*ituA::mls*), and 459 (*fenC::spc*)), we found that fengycin and iturin were required for strain Y6 to antagonize the rice sheath blight pathogen. This was consistent with previous studies reporting that surfactin exhibits significant antibacterial activity and that fengycin and iturin have activities against fungal pathogens [26,27,28]. Our results showed that iturin had stronger antagonistic activity than fengycin against *Rhizoctonia solani*. Zhang et al. reported that iturins produced by *B. velezensis* Jt84 play a key role in rice blast disease biocontrol [29].

Transcriptome analysis of the root system of rice pot trials was used in this experiment to further elucidate the biological control mechanism of *B. velezensis* Y6 against rice sheath blight. According to GO analysis, DEGs in rice roots after inoculation with *B. velezensis* Y6 mainly involved BP aspects of response to stimulus (GO:0050896), response to stress (GO:0006950), and response to abiotic stimulus (GO:0009628). Hao et al. reported that the DEGs in the maize were enriched in eight biological process terms under drought stress in maize, which were mainly related to stress response (e.g., “response to stimulus”, “response to stress”, “response to abiotic stimulus”, “response to biotic stimulus”, “response to endogenous stimulus”, and “response to external stimulus”) [30]. Bollier et al. reported that most DEGs in tomato response to heat stress belonged to GO classes linked to response to stimulus (GO:0050896), response to stress (GO:0006950) and response to abiotic stimulus (GO:0009628) [31]. Liang et al. reported that the DEGs in rice were enriched in the categories of response to temperature stimulus (GO:0009266), response to abiotic stimulus (GO:0009628), response to heat (GO:0009408), and response to stimulus (GO:0050896) under mild field drought stress during grain-filling stage [32]. Yan et al. reported that the DEGs in tomato treated with *B. velezensis* YYC were enriched in the categories of defense response (53 DEGs), response to fungus (20 DEGs) and defense response to fungus (19 DEGs) [12]. Usually, stresses such as high temperature, drought, and pathogens will affect the growth of rice and cause a decrease in rice yield [33,34,35]. Our results showed that inoculation with strain Y6 could increase rice yield in the field. The results of this study are consistent with current research, indicating that inoculation with strain Y6 can improve rice resistance to stress.

It has been reported that the phenylpropanoid pathway was crucial in the defense against plant diseases. When the cell wall is invaded by pathogens, the phenylpropanoid pathway is greatly activated, providing lignin-building monolignols, thus forming the first line of defense for plants against diseases [36,37]. Vranic et al. reported that activation of the wheat phenylpropanoid pathway can prevent fungal invasion and spread through the spike, thereby enhancing wheat plant resistance to Fusarium head blight [38]. According to the study by Liu et al., strain YC0136 can enhance systemic resistance in tobacco. After inoculation with strain YC0136, the expression of certain genes involved in the phenylpropanoid metabolic pathway in tobacco roots showed varying degrees of increase compared with tobacco roots that were not inoculated with strain YC0136 [39]. Graham et al. reported that glutathione is a key regulator of plant defense and a crucial player in governing the outcome of biotic stresses [40]. In this study, we found that the DEGs were involved in phenylpropanoid biosynthesis (map00940), glutathione metabolism (map00480), glycolysis/gluconeogenesis (map00010), and amino sugar and nucleotide sugar metabolism (map00520), which indicates that inoculation with *B. velezensis* Y6 can enhance the metabolic pathways related to plant defense in rice.

## 5. Conclusions

In this study, we found that *B. velezensis* Y6 could significantly inhibit the growth of mycelium in *Rhizoctonia solani*. Lipopeptides play an important role in the control of rice sheath blight by *B. velezensis* Y6, among which iturin and fengycin are essential, and iturinW, a novel lipopeptide in *B. velezensis* Y6, plays a major role in lipopeptide antagonism to *Rhizoctonia solani*. We also found that inoculation with *B. velezensis* Y6 can increase rice yield in the field. In addition, transcriptome analysis showed that strain Y6 may improve rice resistance to *Rhizoctonia solani*, mainly by upregulating genes related to response to stress, response to abiotic stimulus, phenolylpropanoid biosynthesis, and glutathione metabolism. This study helps to reveal the molecular mechanism of *B. velezensis* Y6 in the biological control of rice sheath blight and provides a theoretical basis for the application of *B. velezensis* Y6 in the biological control of rice sheath blight.

## Figures and Tables

**Figure 1 microorganisms-12-01694-f001:**
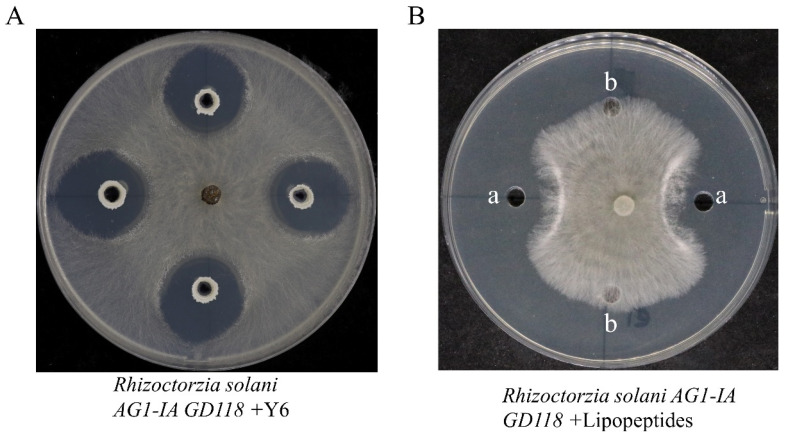
Inhibitory effect of *B. velezensis* Y6 and lipopeptides against *Rhizoctonia solani* mycelium growth. (**A**) Photograph of the inhibitory effect of *B. velezensis* Y6 on the mycelial growth of *Rhizoctonia solani*; (**B**) photograph of the inhibitory effect of lipopeptides on the mycelial growth of *Rhizoctonia solani*, (a) crude lipopeptides, (b) methanol.

**Figure 2 microorganisms-12-01694-f002:**
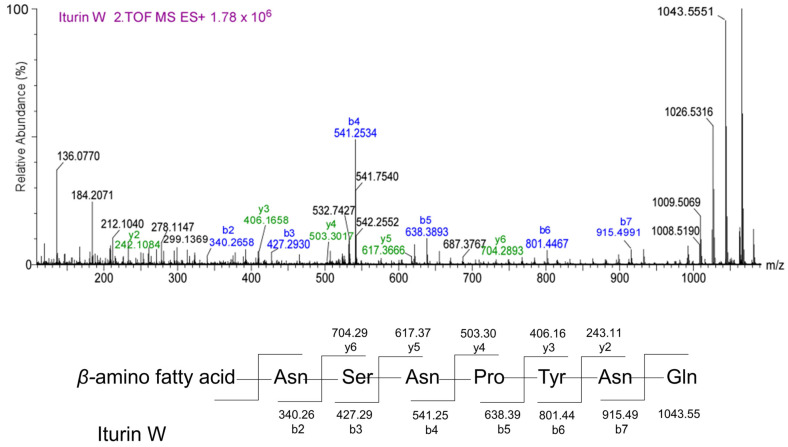
MS/MS analysis of iturin. MS/MS spectra of the iturin produced by *B. velezensis* Y6 and possible b- and y-type fragments generated from iturin W by collision.

**Figure 3 microorganisms-12-01694-f003:**
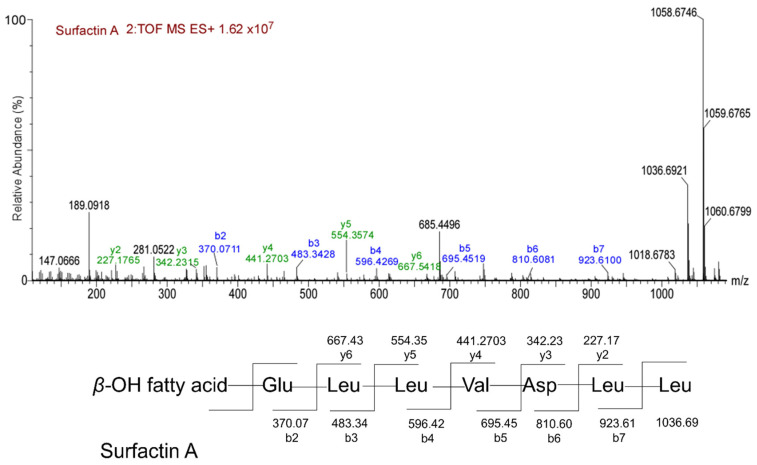
MS/MS analysis of surfactin. MS/MS spectra of the surfactin produced by *B. velezensis* Y6 and possible b- and y-type fragments generated from surfactin A by collision.

**Figure 4 microorganisms-12-01694-f004:**
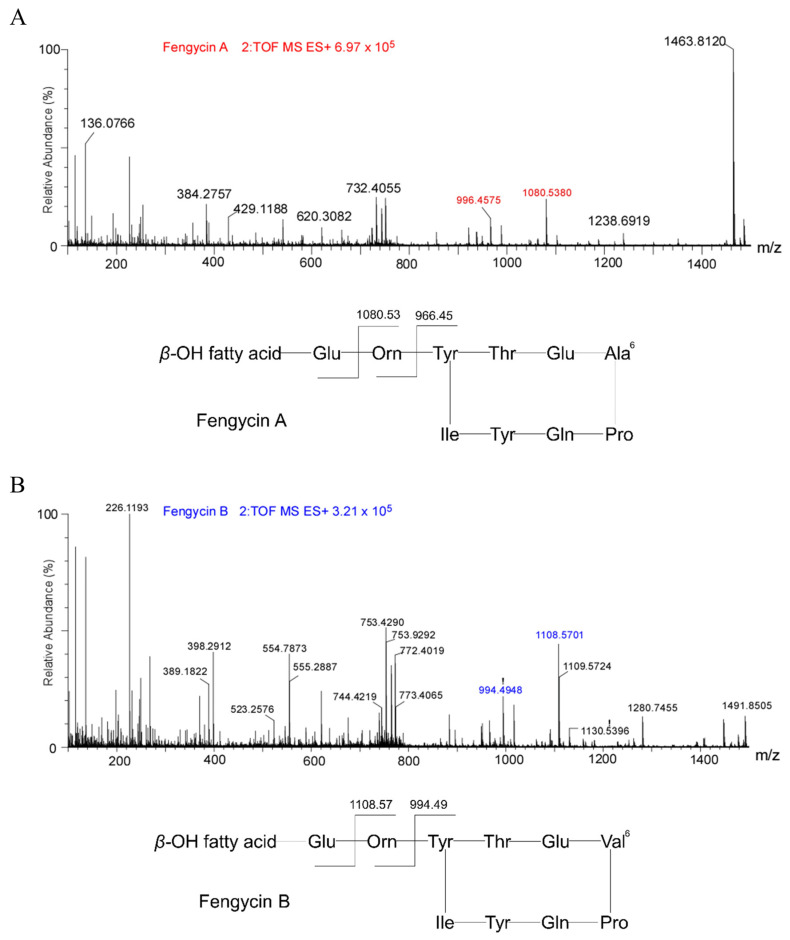
MS/MS analysis of fengycin. (**A**) MS/MS spectra of the peak at *m*/*z* of 1463.8120 fengycin produced by *B. velezensis* Y6 and possible basic structures of Fengycin A; (**B**) MS/MS spectra of the peak at *m*/*z* of 1491.8505 fengycin produced by *B. velezensis* Y6 and possible basic structures of Fengycin B.

**Figure 5 microorganisms-12-01694-f005:**
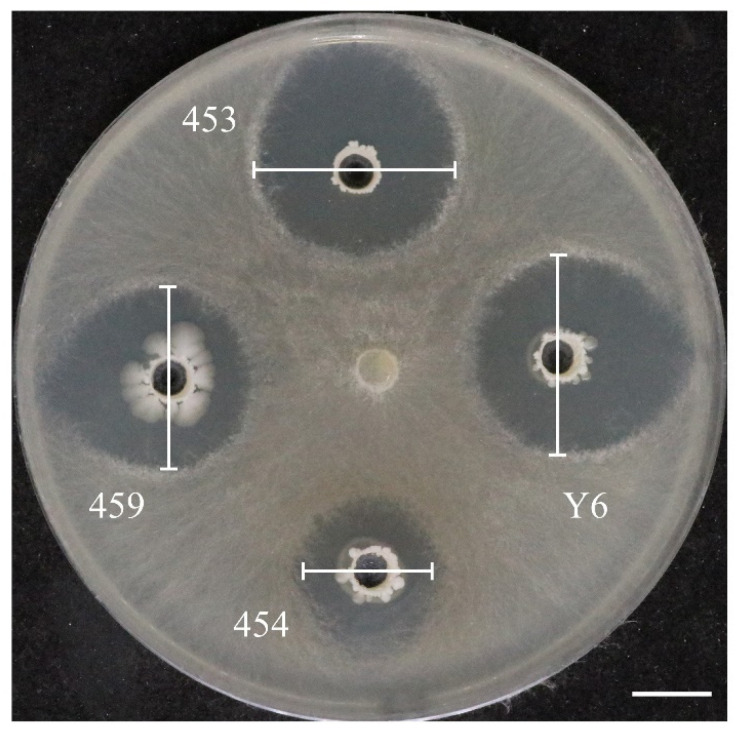
The antagonistic activity of strain Y6 and its mutants (453 (*srfAA::mls*), 454 (*ituA::mls*), and 459 (*fenC::spc*)). The white line represents the size of the inhibition circle, which was evaluated after 24 h of incubation at 30 °C. The bar symbolizes 10 mm.

**Figure 6 microorganisms-12-01694-f006:**
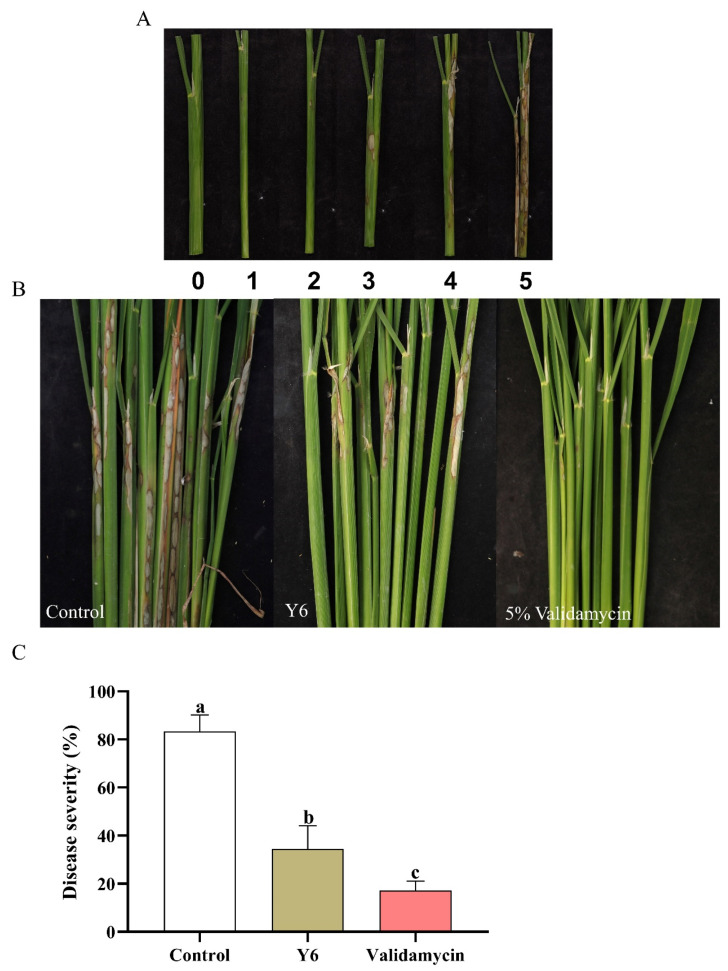
*B. velezensis* Y6 can reduce the disease severity of rice sheath blight in pot experiments. (**A**) Photographs of rice sheath blight severity rating values. (**B**) Photographs of rice sheath blight under different treatments. (**C**) Disease severity of rice sheath blight. Different letters indicate significant differences.

**Figure 7 microorganisms-12-01694-f007:**
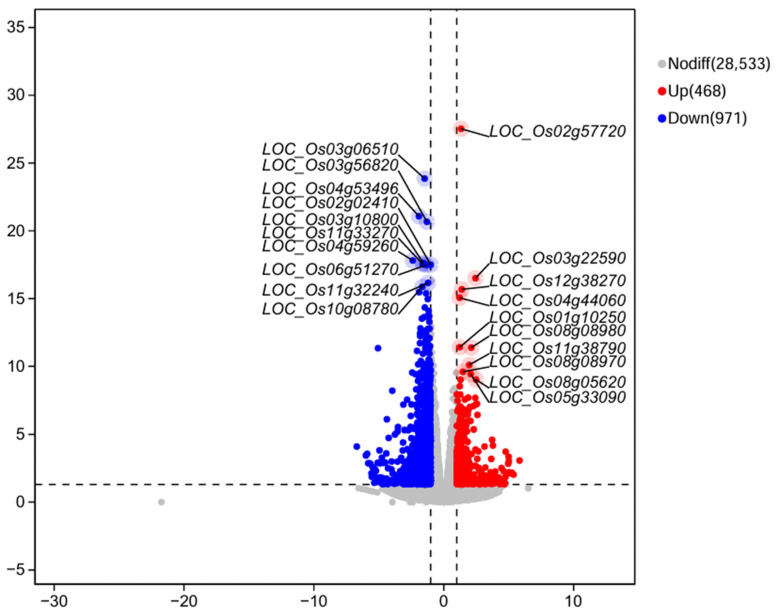
Volcano plot of differentially expressed genes (DEGs) in the transcriptome of rice inoculated with *B. velezensis* Y6 and control. Red dots indicate up-regulated genes, and blue dots indicate down-regulated genes.

**Figure 8 microorganisms-12-01694-f008:**
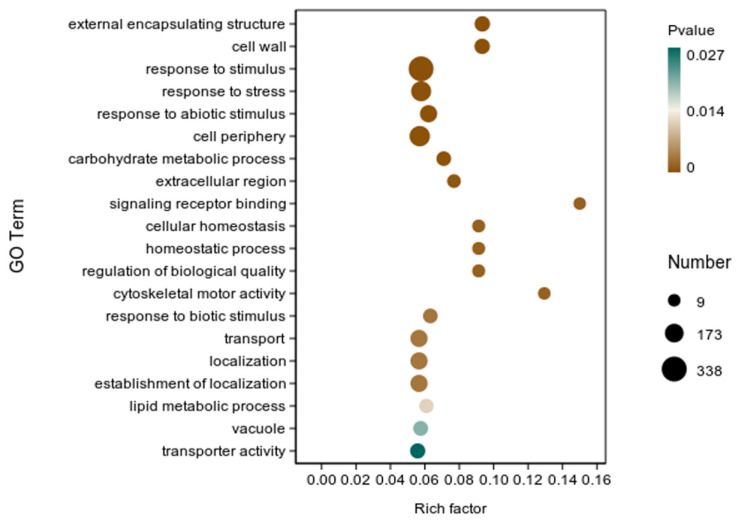
GO enrichment analysis of differentially expressed genes (DEGs) in the transcriptome of rice inoculated with strain Y6 and control. The X-axis represents the enrichment factor. The Y-axis represents the GO term name.

**Figure 9 microorganisms-12-01694-f009:**
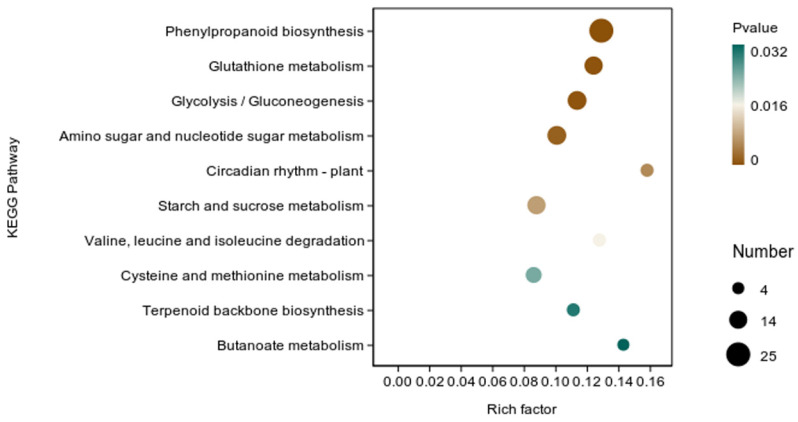
KEGG enrichment analysis of differentially expressed genes (DEGs) in the transcriptome of rice inoculated with *B. velezensis* Y6 and control. The X-axis represents the enrichment factor. The Y-axis represents the pathway name.

**Table 1 microorganisms-12-01694-t001:** UPLC/Q-TOF Time-of-Flight Mass Spectrometer detection mobile phase settings.

Time (min)	Mobile Phase A%	Mobile Phase B%	Flow Rate
4.5	80	20	0.3 mL/min
5	80	20	0.3 mL/min
7	95	5	0.3 mL/min
9	95	5	0.3 mL/min
10	40	60	0.3 mL/min
15	40	60	0.3 mL/min

**Table 2 microorganisms-12-01694-t002:** The effects of *B. velezensis* Y6 and its derived mutants on the mycelial growth of *Rhizoctonia solani*.

Strain	Diameter of the Inhibition Zone (mm)
WT (Y6)	24.88 ± 0.22 a
453 (*srfAA::mls*)	24.50 ± 0.50 a
454 (*ituA::mls*)	17.13 ± 0.74 c
459 (*fenC::spc*))	21.88 ± 0.65 b

*Note:* Different letters indicate significant differences.

## Data Availability

The data presented in the study are deposited in the NCBI repository, accession number PRJNA1136222.

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
