# Peer review of "Bacillus velezensis Y6, a Potential and Efficient Biocontrol Agent in Control of Rice Sheath Blight Caused by Rhizoctonia solani"

_microorganisms, 2024, doi:10.3390/microorganisms12081694_

Round 1

Reviewer 1 Report

Comments and Suggestions for Authors

In the strains it mentions some B. velezensis mutants, however, it does not refer to what mutated in these strains or why they are being evaluated.

The authors should include a table with the percentages of inhibition of mycelial growth of the wild strain and mutants.

 Why did they only evaluate the wild strain B. velezensis and not the mutants in rice plant tissues?

Comments on the Quality of English Language

Minor editing of English language required

Author Response

  1. Response to comment: In the strains it mentions some B. velezensis mutants, however, it does not refer to what mutated in these strains or why they are being evaluated.

Response: Considering the Reviewer’s suggestion, we added information about the mutant strains in the article. ituA srfAA and fenC are the key genes for lipopeptide synthesis. The loss of these genes will result in the inability to synthesize iturin surfactin and fengycin respectively. Mutants deficient in lipopeptide synthesis were generated by replacing the coding region with a resistance cassette using long flanking homology PCR (LFH-PCR) followed by DNA transformation as previously described.

  1. Response to comment: The authors should include a table with the percentages of inhibition of mycelial growth of the wild strain and mutants.

Response: As the reviewer suggested, we have added Table 2 to the paper.

  1. Response to comment: Why did they only evaluate the wild strain B. velezensis and not the mutants in rice plant tissues?

Response: Our experiment aims to evaluate the control effect of B. velezensis Y6 on rice sheath blight. The article's experimental results can demonstrate the control effect. Mutant strains need to be in the presence of antibiotics to express corresponding traits. In a pot test, antibiotics on the surface of plant tissues are easily inactivated, and there is no guarantee whether the traits of the mutant strains will change.

Special thanks to you for your good comments.

Reviewer 2 Report

Comments and Suggestions for Authors

The need to obtain ever greater yields and the associated need to use plant protection products have become a significant problem in recent decades. Excessive use of plant protection products is associated not only with environmental pollution resulting in a decrease in biodiversity but also with an increase in pathogen resistance to these products. The reviewed studies are among those that deal with the possibility of replacing the use of chemical agents with biological control. In this sense, the reviewed studies can provide not only theoretical but above all practical knowledge.

Lines 104-116, 2.Field Experiment: please provide information on how many plots there were in each of the experimental variants. This is not clearly stated in the methods.

Line 284, Figure 8 : it would be worth presenting this Figure earlier, in the methods as an explanation of the experimental scheme

Line 324: it would be a good idea to explain the abbreviations CC, BP, MF in the caption under the Figure, it would make things easier for the reader

Author Response

  1. Response to comment: Lines 104-116,2. Field Experiment: please provide information on how many plots there were in each of the experimental variants. This is not clearly stated in the methods.

Response: We have added the following information to the methods: Each treatment was repeated four times.

  1. Response to comment: Line 284, Figure 8: it would be worth presenting this Figure earlier, in the methods as an explanation of the experimental scheme

Response: As the reviewer suggested, Figure 8 deserves to be shown earlier, but the results shown in the previous figures are logically continuous. So, I think Figure 8 is also reasonable in its current position. Thank you again for your suggestion.

  1. Line 324: it would be a good idea to explain the abbreviations CC, BP, MF in the caption under the Figure, it would make things easier for the reader

Response: Indeed, as suggested by the reviewer, we have added explanations for the abbreviations CC, BP, and MF in the captions below the figures.

Special thanks to you for your good comments.